# Effect on growth of exposure to maternal antiretroviral therapy in breastmilk versus extended infant nevirapine prophylaxis among HIV-exposed perinatally uninfected infants in the PROMISE randomized trial

Lynda Stranix-Chibanda[1,2]*, Camlin Tierney[3], Mauricio Pinilla[3], Kathleen George[4], Jim Aizire[5,6], Godwin Chipoka[7], Macpherson Mallewa[8], Megeshinee Naidoo[9], Teacler Nematadzira[2], Bangani Kusakara[2], Avy Violari[10], Tapiwa Mbengeranwa[2], Boniface Njau[11], Lee Fairlie[12], Gerard Theron[13], Mwangelwa Mubiana-Mbewe[14], Sandhya Khadse[15], Renee Browning[16], Mary Glenn Fowler[17], George K. Siberry[18], for the PROMISE Study Team¶

**1** University of Zimbabwe Faculty of Medicine and Health Sciences, Child and Adolescent Health Unit, Harare, Zimbabwe, **2** University of Zimbabwe Clinical Trials Research Centre, Harare, Zimbabwe, **3** Harvard T.H. Chan School of Public Health, Center for Biostatistics in AIDS Research in the Department of Biostatistics, Boston, MA, United States of America, **4** FHI 360, Durham, NC, United States of America, **5** Makerere University—Johns Hopkins University Research Programme, Kampala, Uganda, **6** Johns Hopkins Bloomberg School of Public Health, Baltimore, MD, United States of America, **7** University of North Carolina Project, Lilongwe, Malawi, **8** College of Medicine—Johns Hopkins University Project, Blantyre, Malawi, **9** University of KwaZulu-Natal, Centre Aids Prevention Research South Africa (CAPRISA), Durban, South Africa, **10** Perinatal HIV Research Unit, Johannesburg, South Africa, **11** Kilimanjaro Christian Medical Center, Moshi, United Republic of Tanzania, **12** Wits Reproductive Health and HIV Institute, University of the Witwatersrand, Johannesburg, South Africa, **13** Department of Obstetrics and Gynaecology, Stellenbosch University, Cape Town, South Africa, **14** Centre for Infectious Disease Research in Zambia, Lusaka, Zambia, **15** Department of Obstetrics and Gynaecology, BJ Government Medical College, Pune, India, **16** Division of AIDS, National Institute of Allergy and Infectious Diseases, Bethesda, MD, United States of America, **17** Johns Hopkins University School of Medicine, Baltimore, MD, United States of America, **18** Office of HIV/AIDS, United States Agency for International Development, Washington, DC, United States of America

¶ Complete membership of the author group can be found in the Acknowledgments
* lstranix@uz-ctrc.org, stranixchibanda@gmail.com

**Data Availability Statement:** The individual level data cannot be made publicly available due to the

## Abstract

### Background

Malnutrition is highly prevalent in HIV-exposed perinatally uninfected infants (HEUs) increasing the risk of morbidity and mortality throughout the life course. We set out to compare the effect of postnatal exposure to maternal antiretroviral therapy (mART) in breastmilk versus infant Nevirapine prophylaxis (iNVP) on somatic growth of HEUs in the randomized PROMISE trial.

### Methods and findings

We randomized 2431 mothers with HIV and their 2444 HEUs from six African countries and India 6–14 days after delivery to mART or iNVP for prevention of breastmilk HIV

ethical restrictions in the study's informed consent documents and in the International Maternal Pediatric Adolescent AIDS Clinical Trials (IMPAACT) Network's approved human subjects protection plan; public availability may compromise participant confidentiality. However, data are available to all interested researchers upon request to the IMPAACT Statistical and Data Management Centre's data access committee by email to sdac. data@fstrf.org or sdac.data@sdac.harvard.edu. This committee reviews and responds to requests for data, obtains necessary approvals from IMPAACT leadership and the NIH, arranges for signature of a Data Use Agreement, and releases the requested data.

**Funding:** Overall support for the International Maternal Pediatric Adolescent AIDS Clinical Trials Network (IMPAACT) was provided by the National Institute of Allergy and Infectious Diseases (NIAID) with co-funding from the Eunice Kennedy Shriver National Institute of Child Health and Human Development (NICHD) and the National Institute of Mental Health (NIMH), all components of the National Institutes of Health (NIH), under Award Numbers UM1AI068632 (IMPAACT LOC), UM1AI068616 (IMPAACT SDMC) and UM1AI106716 (IMPAACT LC), and by NICHD contract number HHSN275201800001I. The content is solely the responsibility of the authors and does not necessarily represent the official views of the NIH.

**Competing interests:** The authors have declared that no competing interests exist.

transmission. The mART regimen contained tenofovir/emtricitabine (99%) plus lopinavir/ritonavir. Infant growth parameters were compared at postnatal week 10, 26, 74 and 104 using World Health Organization (WHO) z-scores for length-for-age (LAZ), weight-for-age (WAZ), and head circumference-for-age (HCAZ). Week 26 LAZ was the primary endpoint measure. Student T-tests compared mean LAZ, WAZ, and HCAZ; estimated mean and 95% confidence interval (CI) are presented.

Maternal and infant baseline characteristics were comparable between study arms. The estimated median breastfeeding duration was 70 weeks. After a mean follow-up of 88 weeks, mean LAZ and WAZ were below the WHO reference population mean at all time-points, whereas mean HCAZ was not. The mART and iNVP arms did not differ for the primary outcome measure of LAZ at week 26 (p-value = 0.39; estimated mean difference (95%CI) of -0.05 (-0.18, 0.07)) or any of the other secondary growth outcome measures or timepoints (all p-values≥0.16). Secondary analyses of the primary outcome measure adjusting for week 0 LAZ and other covariates did not change these results (all p-values≥0.09). However, infants assigned to mART were more likely to have stunting compared to iNVP infants at week 26 (odds ratio (95% CI): 1.28 (1.05, 1.57)).

## Conclusions

In HEUs, growth effects from postnatal exposure to mART compared to iNVP were comparable for measures on length, weight and head circumference with no clinically relevant differences between the groups. Despite breastfeeding into the second year of life, length and weight were below reference population means at all ages in both arms. Further investment is needed to optimize postnatal growth of infants born to women with HIV.

## Clinical trial registration

ClinicalTrials.gov number NCT01061151.

## Introduction

Prior research suggests that HIV-exposed uninfected infants (HEUs) have diminished somatic growth compared to unexposed infants [1–5]. Wasting, stunting and underweight more than double the hazard of childhood mortality [6–8] while their early life effect on neurocognition diminishes developmental potential in adult life [9]. The relative contributions of advanced maternal HIV disease, restricted breastfeeding and antiretroviral (ARV) drug exposures are not clear, leading to mixed conclusions in published literature about the effect of *in utero* ARV exposure on birth size and postnatal growth [10–12]. Previous guidelines for antiretroviral treatment (ART) reserved treatment for pregnant women with advanced HIV disease, itself a predictor of poor infant growth. Similarly, earlier recommendations to avoid or limit the duration of breastfeeding for infants exposed to HIV complicated evaluations of growth in infant populations where early cessation of breastfeeding or not breastfeeding increases infant mortality and morbidity, mostly in resource limited settings [13,14]. The current recommendation in such settings is to provide life-long ART to pregnant women with HIV and to breastfeed [15], with many countries discouraging weaning before 2 years. The effect on growth of

exposure to maternal ART (mART) throughout the first 1000 days of a child's life (from conception through second birthday) is not well established.

The postpartum randomization in the multi-country PROMISE trial provided an ideal opportunity to study postnatal growth in breastfed infants exposed to mART but not to advanced maternal HIV disease. Primary analyses of the relative safety and efficacy of the ARV strategies evaluated were previously published [16–18]. This secondary analysis assesses the effect of breastmilk exposure to mART on infant/child growth through 104 weeks of life compared to extended infant Nevirapine prophylaxis (iNVP), assessed by length-for-age (LAZ), weight-for-age (WAZ) and head circumference-for-age (HCAZ) z-scores.

## Methods

### Study design and participants

PROMISE was a randomized open-label trial to evaluate the relative efficacy and safety of several proven ARV strategies to prevent vertical HIV transmission and optimize maternal health and child survival among healthy women who did not meet local criteria to initiate ART in settings with varying standards of care [13–15]. Fig 1 presents the overall PROMISE trial schema in countries where maternal ART was not standard at the time, illustrating the three sequential PROMISE components; Antepartum, Postpartum and Maternal Health (S1 Fig). The Postpartum randomization for breastfeeding mother-infant pairs was conducted at 14 health facility-based research sites in seven countries: India, Malawi, South Africa, Tanzania, Uganda, Zambia and Zimbabwe.

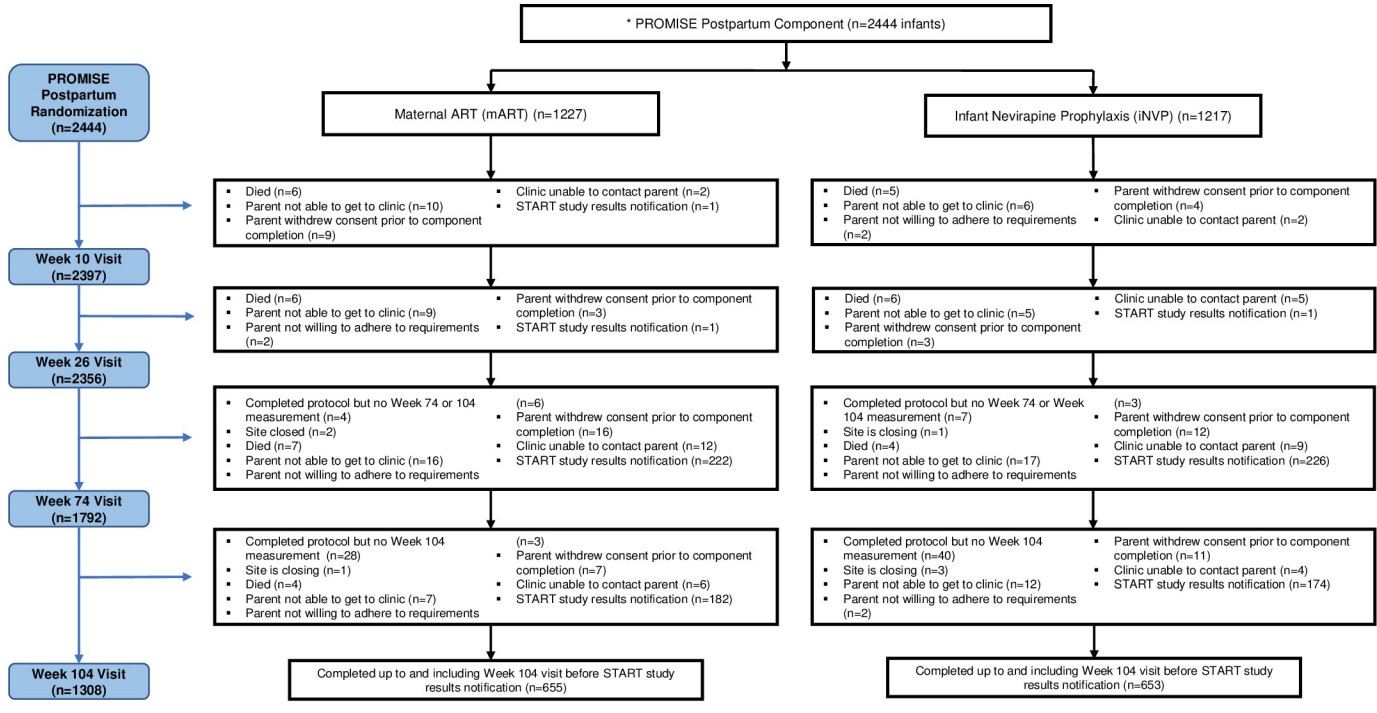

*Abbreviations*: iNVP—infant nevirapine prophylaxis; mART—maternal antiretroviral treatment; n—number

**Fig 1. Participant flow diagram.** There was no apparent difference in premature study discontinuation for reasons other than death; 23 in the mART arm (2% of 1,227 randomized) and 15 in the iNVP arm (1% of 1217) (p = 0.81). The median (Q1-Q3) follow-up in both arms was to 104 weeks of age (74–105), with a mean of 88 weeks. Most infants still breastfed (93.7% (95%CI: 92.6%, 94.6%)) at 26 weeks of age. The estimated median (Q1-Q3) duration of breastfeeding and, hence, postnatal ARV exposure was 70 weeks (56–82) and not different in both study arms. As previously reported [16], 14 breastfed infants acquired HIV infection during postnatal follow-up (7/1219 mother-infant pairs in the maternal ART arm and 7/1211in the infant NVP arm).

Women were recruited with their uninfected infants to the Postpartum Component between 2011 and 2015 after completing the Antepartum Component or when first identified as HIV-infected during active labor ("late presenters"). Maternal inclusion criteria included the intent to breast feed, a CD4 count >350 cells/mm$^3$ at enrollment and above the country specific threshold for initiation of ART, hemoglobin ≥7.0 grams/dL, white blood cell count (WBC) ≥1500 cells/mm$^3$, absolute neutrophil count (ANC) ≥750 cells/mm$^3$, platelets ≥50,000 cells/mm$^3$, alanine aminotransferase (ALT) ≤2.5x upper limit of normal, and estimated creatinine clearance of ≥60mL/min (Cockcroft-Gault equation for women). Inclusion criteria for infants included age ≤14 days, negative HIV nucleic acid test on a specimen drawn prior to the week 1 visit, hemoglobin ≥10 grams/dL, WBC ≥1,500 cells/mm$^{3,}$ ANC ≥750 cells/mm$^3$, platelets ≥50,000 cells/mm$^3$, and ALT ≤2.5x ULN. Mothers were excluded if they required ART for their own health. For multiple births, a mother-infant pair was enrolled only if all infants could be enrolled. Infants were excluded for life-threatening conditions or circumstances that would make long-term follow-up unlikely in the judgment of the site physician or when birthweight was <2000g.

## Randomizations

*In utero* ARV exposure for infants in the Postpartum Component differed by duration and regimen. At study entry in pregnancy, women with their unborn infant(s) were randomized to receive either prophylaxis with Zidovudine (plus single dose Nevirapine and a 1-week tail of Tenofovir/Emtricitabine) or triple ART (with Zidovudine/Lamivudine plus ritonavir boosted Lopinavir or Tenofovir/Emtricitabine plus ritonavir boosted Lopinavir) from the time of registration at variable gestational ages. Women presenting late for antenatal care in labor or post-delivery received no prior ARVs.

Mother-infant pairs were randomized in the Postpartum Component in a 1:1 ratio at 6–14 days post-delivery to mART with Tenofovir/Emtricitabine plus (preferred) Lopinavir-ritonavir or to iNVP using age-based daily NVP dosing (Fig 1). A web-based, central computer randomization system on the IMPAACT Data Management Center's portal was used. The system used permuted block allocation with stratification by country and the antepartum/intrapartum maternal ARV prophylaxis. A site staff member enrolled participants by answering eligibility and stratification questions in the randomization system, and if passed, the system randomized the participant and generated a coded study identification number which the site pharmacist compared to a master list and dispensed the corresponding study drug.

In the Postpartum Component, the randomly assigned postpartum ARV regimen (mART or iNVP) was continued for the duration of breastfeeding plus 42 days (two weeks following breastfeeding cessation, defined as no breastmilk exposure for >28 days) or through age 18 months, whichever came first, unless stopped for infant HIV infection, toxicity or other medical reason. All study infants received standard-of-care iNVP during the first six weeks of life and cotrimoxazole prophylaxis thereafter, per local guidelines.

On July 6, 2015 PROMISE sites were notified that all maternal participants should be offered life-long ART for their health based on the results of the Strategic Timing of AntiRetroviral Treatment (START) study [19] which demonstrated a significant benefit to beginning ART early, including in those with high CD4 counts. Analyses are thus based on data collected at visits through the date of notification.

## Growth monitoring

Length, weight and head circumference were measured by trained study personnel using standard methods at birth and weeks 1, 6, 10, 14, 26, 38, 50, 62, 74, 86, 98 and 104 of life. Measurements of length and weight were performed using standardized rigid length boards and

calibrated weighing scales. Mothers received infant feeding counselling throughout the study and were supported to breastfeed exclusively for the first six months of life followed by the gradual introduction of appropriate foods and fluids. The timing of cessation of breastfeeding was determined by the mother. Breastfeeding status and infant illnesses were recorded at each visit. The full PROMISE 1077BF protocol can be accessed on the protocol website at https://www.impaactnetwork.org/studies/1077BF.asp.

## Study oversight

The study was funded by the National Institutes of Health (ClinicalTrials.gov number NCT01061151). Written informed consent was obtained from each maternal study participant for her and her child's participation. Study conduct adhered to international guidelines and was reviewed every six months by an independent Data and Safety Monitoring Board. Ethics committees and institutional review boards that approved this study include—MUJHU/Kampala, Uganda: The Joint Clinical Research Centre (JCRC) IRB, the National Drug Authority and the Uganda National Council of Science and Technology (UNCST) in Uganda and the Johns Hopkins Medical Institutions (JHMI) IRB in the U.S.; Wits RHI Shandukani CRS and Soweto IMPAACT CRS, Johannesburg, South Africa: University of Witwatersrand Human Ethics Research Committee (Medical), Medicines Control Council (South African Health Products Regulatory Authority in February 2018); FAM-CRU CRS, Cape town, South Africa: Health Research Ethics Committee (HREC), Faculty of Health Sciences, Stellenbosch University and Medicines Control Council (South African Health Products Regulatory Authority in February 2018); Durban Paediatric HIV CRS, Durban, South Africa: University of KwaZulu-Natal (UKZN) Biomedical Research Ethics Committee, Medicines Control Council (South African Health Products Regulatory Authority in February 2018); George CRS, Lusaka, Zambia: University of North Carolina (UNC) at Chapel Hill Biomedical IRB and University of Zambia Biomedical Research Ethics Committee (UNZABREC); Harare, Seke North and St. Mary's sites, Zimbabwe: Medical Research Council of Zimbabwe(MRCZ), Research Council of Zimbabwe (RCZ), Medicine Control Authority of Zimbabwe (MCAZ), Joint Parirenyatwa group of Hospitals/University of Zimbabwe College of Health Sciences Research Ethics Committee(JREC); Byramjee Jeejeebhoy Medical College (BJMC) CRS, Pune, India: BJ Government College CTU Ethics Committee and Johns Hopkins IRB; Blantyre, Malawi: College of Medicine Research and Ethics Committee (COMREC) in Malawi, Pharmacy, Medicines and Poisons Board and Johns Hopkins Medical Institutions (JHMI) IRB in the U.S.; Lilongwe, Malawi: National Health Sciences Research Committee (NHSRC) in Malawi Pharmacy, Medicines and Poisons Board, and University of North Carolina, Chapel Hill (UNC-CH) Office of Human Research Ethics IRB in the U.S and Kilimanjaro Christian Medical Centre (KCMC), Moshi, Tanzania: Kilimanjaro Christian Medical College Ethics Committee, National Health Research Ethics Committee and Tanzania Medicines and Medical Devices Authority.

## Statistical methods

Age and sex-appropriate World Health Organization z-scores were computed for LAZ, WAZ and HCAZ based on the SAS macro for z-score calculations for children aged <61 months [20]. The primary objective comparison timepoint was Week 26 and primary outcome measure was LAZ at Week 26; LAZ at weeks 10, 74, and 104, as well as WAZ and HCAZ at weeks 10, 26, 74, and 104 were secondary outcome measures. Mean z-scores for each anthropometric measure were compared at these timepoints between the assigned study arms—mART versus iNVP. Primary analyses were carried out using an intent to treat approach (analyzed as

randomized), as were secondary analyses on additional growth outcome measures. Selected subgroup and restricted/as-treated analyses were performed as secondary analyses. Except for accrual and baseline summaries the unit of analysis was the infant. Twins were analyzed separately.

Mean age adjusted z-scores were compared with a Student's t-test. Modification of the mART versus iNVP effect by subgroups (interaction tests) were assessed via linear regression. Comparisons of other data applied Wilcoxon/Kruskal-Wallis for continuous data, $X^2$/exact tests for categorical data, as appropriate. Treatment effects for binary outcomes were summarized with odds ratios from logistic regression. P-values and tests were two-sided and nominal with a significance level set at 0.05. The z-scores were also compared for LAZ stunting, WAZ underweight and HCAZ using a binary categorization with the following standard deviation (SD) cut offs: z-score < -2 SD and ≥-2 SD [21].

The individual level data cannot be made publicly available due to the ethical restrictions in the study's informed consent documents and in the International Maternal Pediatric Adolescent AIDS Clinical Trials (IMPAACT) Network's approved human subjects protection plan; public availability may compromise participant confidentiality. However, data are available to all interested researchers upon request to the IMPAACT Statistical and Data Management Centre's data access committee by email to sdac.data@fstrf.org or sdac.data@sdac.harvard.edu. This committee reviews and responds to requests for data, obtains necessary approvals from IMPAACT leadership and the NIH, arranges for signature of a Data Use Agreement, and releases the requested data.

## Results

### Accrual and analysis inclusion

A total of 2431 mother-infant pairs were randomized in the Postpartum Component, with 1220 pairs in the mART and 1211 in the iNVP arms. A total of 128 (5%) pairs had received no ARVs prior to the onset of labor and entered as Late Presenters. For those previously enrolled in the Antepartum Component, 1009 pairs (42%) had been randomly assigned during pregnancy to *in utero* Zidovudine prophylaxis, 1005 pairs (41%) to *in utero* Zidovudine/Lamivudine plus ritonavir boosted Lopinavir and 289 pairs (12%) to Tenofovir/Emtricitabine plus ritonavir boosted Lopinavir.

### Baseline characteristics

Mothers enrolled were predominantly black African women (97%) of median (IQR) age 26.6 years (23.2–30.3) at the time of postpartum randomization and in good health; median (Q1-Q3) BMI 24.7 kg/m$^2$ (22.3–28.0), pre-ART CD4 count 535 cells/mm$^3$ (438–671) and 60% had viral load that was below the lower limit of quantification (36%) or quantifiable and <400 copies/mL (24%). Infants were 51% male, had a median (Q1-Q3) birthweight of 2900g (2600–3200) and gestational age of 39 weeks (38–40). All infants were breastfed from birth. In the mART and iNVP arms respectively, there were seven and six pairs of twins (0.5% of the sample) and 143 infants in each arm (12%) were born prematurely, before 37 completed weeks of gestation. Maternal and infant baseline characteristics were comparable between study arms (Table 1).

### Infant study status

Among the 2444 infants, 816 (33%) were still on-study on July 6, 2015 when sites were notified of the START study results, 1383 (57%) had completed study follow-up or had their site closed (7 (<0.5%) of the 1383) and 245 (10%) had prematurely discontinued follow-up (Fig 1).

**Table 1. Baseline characteristics for maternal and infant participants in the Postpartum Component.**

| | | PP Randomization Arm | |
|---|---|---|---|
| | | **Maternal ART (mART)** | **Infant Nevirapine Prophylaxis (iNVP)** |
| **Maternal Characteristics** | | N = 1220 | N = 1211 |
| Age at randomization (years) | # missing | 0 | 0 |
| | Min-Max | 18.4–43.5 | 18.3–46.6 |
| | Median (Q1-Q3) | 26.7 (23.2–30.4) | 26.5 (23.1–30.3) |
| | Mean (s.d.) | 27.2 (5.2) | 27.0 (5.1) |
| | 10%-90% | 20.8–34.5 | 20.7–34.3 |
| Race | Asian (from Indian subcontinent) | 0 (0%) | 2 (0%) |
| | Black African | 1,178 (97%) | 1,168 (96%) |
| | Indian (Native of India) | 41 (3%) | 40 (3%) |
| | Coloured | 1 (0%) | 1 (0%) |
| AP Randomization/Late Presenter | Late Presenter | 66 (5%) | 62 (5%) |
| | Triple ARV (3TC-ZDV/LPV-RTV) | 508 (42%) | 497 (41%) |
| | Triple ARV (FTC-TDF/LPV-RTV) | 140 (11%) | 149 (12%) |
| | ZDV+sdNVP+TRV tail | 506 (41%) | 503 (42%) |
| Weight (kg) | # missing | 0 | 0 |
| | Min-Max | 35.0–121.9 | 35.4–133.0 |
| | Median (Q1-Q3) | 61.0 (54.9–69.8) | 61.0 (54.7–69.1) |
| | Mean (s.d.) | 63.5 (12.7) | 62.9 (11.9) |
| | 10%-90% | 49.8–80.4 | 49.6–78.3 |
| Height (cm) | # missing | 1 | 0 |
| | Min-Max | 131.4–180.0 | 117–179 |
| | Median (Q1-Q3) | 158.0 (153.0–162.1) | 158.0 (152.3–162.0) |
| | Mean (s.d.) | 157.6 (7.0) | 157.1 (6.9) |
| | 10%-90% | 149–166 | 148.9–165.4 |
| BMI (kg/m$^2$) | # missing | 1 | 0 |
| | Min-Max | 16.6–48.9 | 15.8–50.6 |
| | Median (Q1-Q3) | 24.7 (22.3–28.0) | 24.7 (22.3–28.0) |
| | Mean (s.d.) | 25.5 (4.7) | 25.5 (4.5) |
| | 10%-90% | 20.4–31.9 | 20.4–31.2 |
| Gestational Age at PROMISE entry (weeks) | # missing | 67 | 62 |
| | Min-Max | 13.6–40.1 | 13.7–43.3 |
| | Median (Q1-Q3) | 26.3 (21.4–30.7) | 26.3 (21.7–31.4) |
| | Mean (s.d.) | 26.2 (6.3) | 26.4 (6.2) |
| | 10%-90% | 17.6–35.1 | 17.7–34.6 |
| CD4 at Postpartum screening (cells/mm$^3$) | # missing | 0 | 1 |
| | Min-Max | 351–2019 | 353–2353 |
| | Median (Q1-Q3) | 682.5 (555.0–870.0) | 691 (550–868) |
| | Mean (s.d.) | 734.9 (254.3) | 738.7 (259.0) |
| | 10%-90% | 454–1070 | 457.5–1086.0 |
| HIV RNA level prior to randomization (copies/mL) | Below lower limit of quantification (LLQ) of the assay | 499 (41%) | 379 (31%) |
| | <400 | 276 (23%) | 296 (25%) |
| | 400–1000 | 136 (11%) | 138 (11%) |
| | 1000- <10000 | 194 (16%) | 277 (23%) |
| | 10000 - <100000 | 91 (7%) | 104 (9%) |
| | 100000 - <200000 | 12 (1%) | 9 (1%) |
| | > = 200000 | 10 (1%) | 5 (0%) |

*(Continued)*

**Table 1.** (Continued)

| | | PP Randomization Arm | |
|---|---|---|---|
| | | **Maternal ART (mART)** | **Infant Nevirapine Prophylaxis (iNVP)** |
| WHO stage prior to randomization | Clinical stage I | 1,174 (96%) | 1,182 (98%) |
| | Clinical stage II | 45 (4%) | 28 (2%) |
| | Clinical stage III | 1 (0%) | 1 (0%) |
| **Infant Characteristics** | | **N = 1227** | **N = 1217** |
| Sex | Male | 622 (51%) | 614 (50%) |
| | Female | 605 (49%) | 603 (50%) |
| Gestational Age at Birth (weeks) | # missing | 0 | 0 |
| | Min-Max | 30–44 | 29.6–45.4 |
| | Median (Q1-Q3) | 39 (38–40) | 39 (38–40) |
| | Mean (s.d.) | 39.0 (2.0) | 38.9 (2.0) |
| | 10%-90% | 36–41 | 36–42 |
| Weight at Week 0 visit (0–5 days old) (gm) | # missing | 9 | 6 |
| | Min-Max | 2000–4500 | 2000–4500 |
| | Median (Q1-Q3) | 2910 (2600–3230) | 2900 (2600–3200) |
| | Mean (s.d.) | 2951.1 (442.7) | 2920.3 (434.1) |
| | 10%-90% | 2400–3520 | 2370–3500 |
| Length at Week 0 visit (0–5 days old) (cm) | # missing | 21 | 23 |
| | Min-Max | 37–59 | 36.4–58.0 |
| | Median (Q1-Q3) | 48.6 (47.0–50.0) | 48.0 (46.5–50.0) |
| | Mean (s.d.) | 48.5 (2.9) | 48.1 (2.7) |
| | 10%-90% | 45–52 | 45–51 |
| LAZ at Week 0 visit (0–5 days old) | # missing | 22 | 24 |
| | Min-Max | -6.89–5.19 | -7.00–4.65 |
| | Median (Q1-Q3) | -0.71 (-1.56–0.00) | -0.82 (-1.71–0.03) |
| | Mean (s.d.) | -0.70 (1.51) | -0.90 (1.45) |
| | 10%-90% | -2.42–1.02 | -2.66–0.90 |
| | Not Stunted | 1,018 (84%) | 972 (81%) |
| | Stunted | 187 (16%) | 221 (19%) |
| WAZ at Week 0 visit (0–5 days old) | # missing | 9 | 6 |
| | Min-Max | -3.38–2.48 | -3.38–2.49 |
| | Median (Q1-Q3) | -0.75 (-1.39–0.10) | -0.77 (-1.45–0.13) |
| | Mean (s.d.) | -0.75 (0.98) | -0.82 (0.98) |
| | 10%-90% | -1.94–0.55 | -2.14–0.40 |
| | Not Underweight | 1,101 (90%) | 1,066 (88%) |
| | Underweight | 117 (10%) | 145 (12%) |
| HCAZ at Week 0 visit (0–5 days old) | # missing | 21 | 25 |
| | Min-Max | -7.69–8.67 | -11.82–7.79 |
| | Median (Q1-Q3) | 0.02 (-0.82–0.87) | -0.02 (-0.82–0.79) |
| | Mean (s.d.) | 0.07 (1.30) | -0.09 (1.38) |
| | 10%-90% | -1.41–1.69 | -1.76–1.54 |
| | HCAZ $\geq$-2 | 1,146 (95%) | 1,103 (93%) |
| | HCAZ < -2 | 60 (5%) | 89 (7%) |

*Abbreviations*: AP—Antepartum; ARV—antiretroviral drug; BMI—body mass index; cm—centimeters; FTC—emtracitabine; gm—grams; HCAZ—head circumference-for-age z-score; iNVP—infant nevirapine prophylaxis; kg—kilograms; LAZ—length-for-age z-score; LPV-RTV—ritonavir based lopinavir; m—meters; mART—maternal antiretroviral treatment; mL—milliliters; mm—millimeters; PP—Postpartum; Q1-Q3—interquartile range; s.d.—standard deviation; sdNVP—single-dose nevirapine; TDF—tenofovir diphosphate; TRV—truvada; ZDV—zidovudine; WAZ—weight-for-age z-score; 3TC—lamivudine; 10%-90%—10th to 90th percentile; %—percent.

**Table 2. Summary statistics and comparisons of infant growth measurements by Postpartum Component randomization.**

| | | mART | | iNVP | | | |
|---|---|---|---|---|---|---|---|
| Study Week | Outcome | N | Mean (SE) | N | Mean (SE) | Mean Difference (95% CI) | p-value[a] |
| Week 10 | LAZ | 1158 | -1.19 (0.04) | 1168 | -1.22 (0.04) | 0.02 (-0.10, 0.14) | 0.73 |
| | WAZ | 1157 | -0.37 (0.03) | 1168 | -0.35 (0.03) | -0.02 (-0.11, 0.07) | 0.66 |
| | HCAZ | 1158 | 0.13 (0.04) | 1168 | 0.10 (0.04) | 0.03 (-0.09, 0.14) | 0.63 |
| Week 26 | LAZ (Primary) | 1150 | -0.90 (0.05) | 1162 | -0.85 (0.04) | -0.05 (-0.18, 0.07) | 0.39 |
| | WAZ | 1151 | -0.28 (0.04) | 1161 | -0.28 (0.03) | 0.00 (-0.09, 0.10) | 0.96 |
| | HCAZ | 1151 | 0.60 (0.04) | 1162 | 0.67 (0.04) | -0.07 (-0.18, 0.04) | 0.18 |
| Week 74 | LAZ | 864 | -1.53 (0.05) | 879 | -1.49 (0.04) | -0.04 (-0.16, 0.08) | 0.51 |
| | WAZ | 864 | -0.49 (0.04) | 878 | -0.47 (0.04) | -0.01 (-0.12, 0.09) | 0.82 |
| | HCAZ | 863 | 0.70 (0.05) | 879 | 0.79 (0.05) | -0.09 (-0.22, 0.04) | 0.16 |
| Week 104 | LAZ | 655 | -1.55 (0.05) | 653 | -1.52 (0.05) | -0.03 (-0.17, 0.10) | 0.62 |
| | WAZ | 654 | -0.52 (0.04) | 653 | -0.48 (0.04) | -0.04 (-0.15, 0.08) | 0.53 |
| | HCAZ | 655 | 1.06 (0.06) | 653 | 1.14 (0.06) | -0.08 (-0.26, 0.09) | 0.35 |
| | | mART | | iNVP | | | |
| Study Week | Outcome | N | n (%) | N | n (%) | Odds Ratio (95% CI) | p-value[b] |
| Week 10 | Stunted[†] | 1158 | 322 (27.8) | 1168 | 317 (27.1) | 1.03 (0.86, 1.24) | 0.72 |
| | Underweight[†] | 1157 | 93 (8.0) | 1168 | 67 (5.7) | 1.44 (1.04, 1.99) | 0.03 |
| | HCAZ < -2[†] | 1158 | 89 (7.7) | 1168 | 88 (7.5) | 1.02 (0.75, 1.39) | 0.89 |
| Week 26 | Stunted | 1150 | 258 (22.4) | 1162 | 214 (18.4) | 1.28 (1.05, 1.57) | 0.02 |
| | Underweight[†] | 1151 | 81 (7.0) | 1161 | 63 (5.4) | 1.32 (0.94, 1.85) | 0.11 |
| | HCAZ < -2[†] | 1151 | 19 (1.7) | 1162 | 17 (1.5) | 1.13 (0.58, 2.19) | 0.72 |
| Week 74 | Stunted[†] | 864 | 305 (35.3) | 879 | 281 (32.0) | 1.16 (0.95, 1.42) | 0.14 |
| | Underweight[†] | 864 | 79 (9.1) | 878 | 50 (5.7) | 1.67 (1.15, 2.41) | 0.01 |
| | HCAZ < -2[†] | 863 | 20 (2.3) | 879 | 18 (2.0) | 1.13 (0.60, 2.16) | 0.70 |
| Week 104 | Stunted[†] | 655 | 228 (34.8) | 653 | 215 (32.9) | 1.09 (0.87, 1.37) | 0.47 |
| | Underweight[†] | 654 | 54 (8.3) | 653 | 31 (4.7) | 1.81 (1.14, 2.85) | 0.01 |
| | HCAZ < -2[†] | 655 | 12 (1.8) | 653 | 13 (2.0) | 0.92 (0.42, 2.03) | 0.83 |

Stunted—length-for-age WHO z-score more than 2 standard deviations below reference population mean; Underweight—weight-for-age WHO z-score more than 2 standard deviations below reference population mean; Odds Ratio (95% CI) is from logistic regression

[a] p-value from student's t-test

[b] p-value from Wald $X^2$ test; [†] Post-hoc analysis.

*Abbreviations*: HCAZ—head circumference-for-age z-score; iNVP—infant nevirapine prophylaxis; LAZ—length-for-age z-score; mART—maternal antiretroviral treatment; N—number; SE—standard error; WAZ—weight-for-age z-score; %—percent; 95% CI—95% confidence interval.

## Infant growth outcomes

Summary statistics and comparisons for the infant growth measures and z-scores at the key follow-up time points of weeks 10, 26, 74 and 104 of age are presented in Table 2.

There were no differences between the mART and iNVP arms for the primary outcome measure's mean LAZ at week 26 (p-value = 0.39) or any of the secondary growth outcomes at this time point (all p-values ≥0.18) (Fig 2).

At week 26, in primary analysis, the mean (95% CI) LAZ was -0.90 (-0.99, -0.81) in the mART arm compared to -0.85 (-0.93, -0.76) in the iNVP arm, for a mean difference of -0.05 (-0.18, 0.07). At week 26, the secondary analysis mean WAZ and HCAZ were not different, with mean difference (95% CI) of 0.00 (-0.09, 0.10) in WAZ and -0.07 (-0.18, 0.04) in HCAZ,

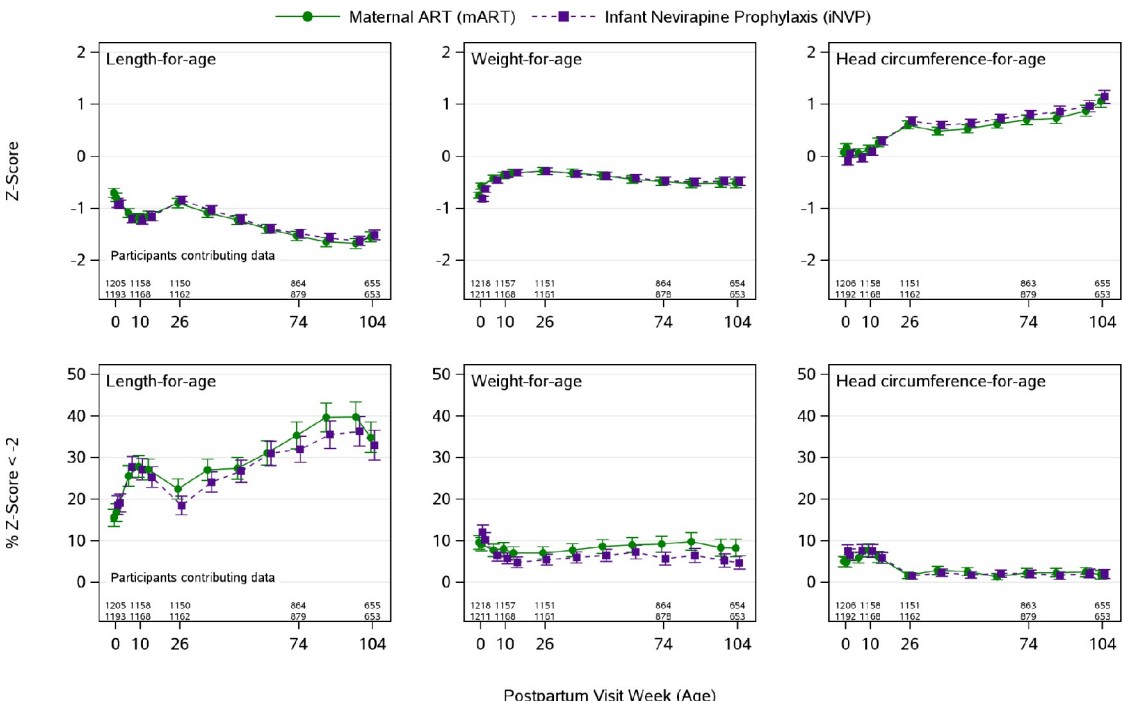

For Participants contributing data, the top and bottom rows refer to the mART and iNVP arm, respectively.
Abbreviations: iNVP—infant nevirapine prophylaxis; mART—maternal antiretroviral treatment; %—percent

**Fig 2. Infant growth outcomes by assigned study arm.**

(p-values = 0.96 and 0.18, respectively). There was no evidence that the postpartum randomization ARV exposure effect on LAZ at week 26 differed by country of residence (p-value = 0.23), *in utero* ARV exposure (p-value = 0.47), or breastfeeding status (p-value = 0.56). Among infants breast feeding at Week 26 the mean (95% CI) difference in LAZ was -0.04 (-0.16, 0.08). At weeks 74 and 104, the mART and iNVP arms did not differ in secondary analyses of mean LAZ, WAZ or HCAZ.

Stunting (LAZ more than two SD below the WHO Child Growth Standards mean in the reference population) at weeks 10, 26, 74 and 104 in the mART arm versus the iNVP arm was measured in 28% vs 27%, 22% vs 18%, 35% vs 32%, and 35% vs 33% infants, respectively. At week 26, infants assigned to the mART arm were more likely to be stunted compared to those randomized to the iNVP arm (OR 1.28 (95%CI 1.05, 1.57), p-value = 0.017, secondary analysis). *Post hoc* analyses revealed no differences in stunting at weeks 10, 74 and 104 with 443/1308 (34%) of infants being stunted at week 104. Underweight scores (WAZ more than two SD below the WHO Child Growth Standards median for that population) at weeks 10, 26, 74 and 104 in the mART arm versus the iNVP arm were observed in 8% vs 6%, 7% vs 5%, 9% vs 6% and 8% vs 5% infants, respectively. In *post hoc* analyses, infants assigned to mART were more likely to be underweight compared to iNVP infants at each of week 10, 74 and 104 (OR (95% CI): 1.44 (1.04, 1.99), 1.67 (1.15, 2.41) and 1.81 (1.14, 2.85), respectively) and at week 26, but the confidence interval includes no difference at week 26 (OR (95% CI): 1.32 (0.94, 1.85)). Consistent trends were seen for LAZ and WAZ in sensitivity analyses using z-score thresholds of -1 and -3 for LAZ and -1 and 2.5 for WAZ (S1 Table). Head circumference measurements revealed no difference in the proportion of infants with HCAZ<-2 between the two study arms (all p-values ≥0.70).

## Discussion

Our data reveal no effect on postnatal infant/child growth of exposure to maternal ART in breastmilk when compared to extended infant NVP prophylaxis in a randomized sample of HIV-exposed and uninfected infants in sub-Saharan Africa and India who breastfed for a median duration of 16 months. These findings are an important addition to the literature in that our analysis from a multi-site randomized clinical trial in resource limited multinational settings examines the effect on growth attributable to maternal ART alone versus extended NVP prophylaxis without added exposure to advanced maternal HIV disease or truncated breastfeeding—a notable gap not addressed in publications before antiretroviral treatment became available [1] and pre-dating the era of universal ART [22,23] when ART was reserved for women with advanced HIV disease. These data are aligned with current literature in the era of universal ART and extended breastfeeding which document suboptimal postnatal growth despite high HIV-free survival when compared to WHO population norms [2] and to HIV unexposed infants [2,5,24].

Our findings extend what is previously known, that administration of daily infant NVP prophylaxis throughout the duration of breastfeeding has no effect on postnatal infant growth [25], and that lower weight at birth attributable to *in utero* maternal ART vs zidovudine exposure corrects rapidly [22]. While stunting at age 6 months was more common in the mART than iNVP arm in our study, the effect was modest and it is unclear whether the effect is a transient effect. Kapito-Tembo *et al* reported lower LAZ at 6 weeks of age in breastfed infants born to pregnant women with HIV on life-long ART from Malawi's Option B+ program but no difference in overall anthropometric growth through 18 months of age when compared to infants not exposed to HIV [26]. The retrospective secondary analysis of the combined Mashi and Mma Bana data sets by Powis *et al* [22] also reported differential mean recumbent length in the first few months of life among more than one thousand singleton infants born HIV-uninfected at term and exposed *in utero* to maternal ART vs zidovudine. While mothers receiving ART in these Botswana cohorts had more advanced HIV disease than those receiving zidovudine alone, the authors rightly call for closer scrutiny of biological explanations for this early but persistent effect on infant length.

Heterogeneity in growth trajectories among breastfed HEUs were described by Lane in Rwanda [2] and Le Roux in South Africa [24]. Characteristics such as maternal height, *in utero* growth restriction, birth weight, birth length and sex were identified as predictive of growth trajectory, most marked for LAZ. In our study, LAZ remained below the WHO standard at all time points and the decline was progressive with age, yet the weight deficit was less marked. As eloquently outlined by Lane [2], maternal ART and extended breastfeeding appear insufficient to sustain normal linear growth of HEUs yet promote adequate weight, resulting in LAZ:WLZ discordance—a combination with unknown impact on long-term health. It is well established that early-life growth and development carry health consequences extending into adulthood, hence the importance of better understanding these observed growth deficits among HEUs in the era of universal ART and extended breastfeeding. Continued study into the underlying mechanisms is needed to inform interventions to address pathways potentially resulting from exposure to maternal HIV and ART toxicities [12].

Major strengths of our data are that the mothers in PROMISE had high CD4 counts with no advanced HIV disease—almost all were WHO Clinical Stage I. Importantly, maternal ART exposure during pregnancy was randomly assigned, thus decreasing selection bias and minimizing unequal distribution of potential confounders like fetal growth or maternal HIV exposure. Also, breastfeeding duration extended into the second year of life, thus mirroring current infant feeding practice in these settings. However, there were some limitations to the study in

that infants included in this analysis excluded those with birth weights below 2000g and the protease inhibitor-based maternal ART regimen assigned in the PROMISE study is not currently widely used. Thus, our findings cannot be generalized to infants of women with HIV receiving other ART regimens or to those with more advance HIV disease than the mothers enrolled in PROMISE. Additionally, our infant follow-up ended at week 104, so we are not able to assess possible longer-term effects of these exposures.

## Conclusion

Administering universal ART to breastfeeding mothers with HIV does not appear to have a negative effect on their infants' growth compared to extended infant NVP prophylaxis in follow up through age 104 weeks, although both groups demonstrated decrements in height relative to WHO norms. While resulting in high infant HIV-free survival, initiating maternal ART and extending breastfeeding still falls short of achieving optimal child health for young children living in resource limited international settings and exposed to HIV. Further research is needed to identify the subsets of HEUs at risk of malnutrition, associated factors and strategies to improve infant growth.

## Supporting information

**S1 Checklist. CONSORT 2010 checklist of information to include when reporting a randomised trial**[*].
(DOC)

**S1 Fig. PROMISE trial schema.**
(TIF)

**S1 Table. Sensitivity analysis comparisons of infant growth measurements by Postpartum Component randomization.**
(DOCX)

**S1 PROMISE study protocol.**
(PDF)

## Acknowledgments

The PROMISE study team gratefully acknowledges the dedication and commitment of the mother-infant pairs and site study teams without whom this study would not have been possible. The content is solely the responsibility of the authors and does not necessarily represent the official views of the NIH, USAID or the United States government. Antiretrovirals were provided free of charge for the PROMISE study by AbbVie, Gilead Sciences, and GlaxoSmithKline.

**PROMISE Study Team Members**: Judith Currier, Katherine Luzuriaga, Adriana Weinberg, James McIntyre, Tsungai Chipato, Karin Klingman, Renee Browning, Mireille Mpoudi-Ngole, Jennifer S. Read, George Siberry, Heather Watts, Lynette Purdue, Terrence Fenton, Linda Barlow-Mosha, Mary Pat Toye, Mark Mirochnick, William B. Kabat, Benjamin Chi, Marc Lallemant, Karin Nielsen; Statistical and Data Analysis Center, Harvard T.H. Chan School of Public Health: Kevin Butler MS, Konstantia Angelidou MS, David Shapiro PHD, and Sean Brummel Ph.D. IMPAACT Operations Center: Anne Coletti, Veronica Toone, Megan Valentine, Kathleen George; Frontier Science Data Management Center: Amanda Zadzilka, Michael Basar, Amy Jennings, Adam Manzella.

**INDIA**. *Byramjee Jeejeebhoy Government Medical College CRS*, *Pune*: Sandesh Patil, MBBS; Ramesh Bhosale, MD; Neetal Nevrekar, MD.

**MALAWI**. *College of Medicine—Johns Hopkins University Project*, *Blantyre*: Salome Kunje, BSc; Alex Siyasiya, Certificate in Microbiology; Mervis Maulidi, Certificate in Nursing and Midwifery. *University of North Carolina Project*, *Lilongwe*: Francis Martinson, MBChB; Ezylia Makina, RNM; Beteniko Milala, BAE.

**SOUTH AFRICA**. *University of KwaZulu-Natal*, *Centre Aids Prevention Research South Africa (CAPRISA)*, *Durban*: Nozibusiso Rejoice Skosana, BN; Sajeeda Mawlana, MBChB. *Family Clinical Research Unit*, Cape Town: Jeanne Louw MSc; Magdel Rossouw MNutr, MBChB; Lindie Rossouw MBChB. *Shandukani CRS*, *Johannesburg*: Masebole Masenya, MD; Janet Grab, BPharm. *Perinatal HIV Research Unit*, *Soweto*: Nasreen Abrahams, MBA; Mandisa Nyati, MBChB; Sylvia Dittmer, MBChB. *Umlazi CRS*, *Durban*: Dhayendre Moodley, MSc, PhD; Vani Chetty, BScHon; Alicia Catherine Desmond, MPharm.

**TANZANIA**. *Kilimanjaro Christian Medical Centre CRS*: Boniface Njau, MPH; Cynthia Asiyo, Bsc; Pendo Mlay, MD.

**UGANDA.** *MU-JHU Research Collaboration CRS*, *Kampala*: Maxensia Owor MBChB, Mmed, MPH; Jim Aizire, MBChB, MHS, PhD; Moreen Kamateeka, MBChB, MPH; Dorothy Sebikari MBChB, MPH.

**ZAMBIA.** *George Clinic CRS*, *Lusaka*: Felistas M. Mbewe, RN, BSc; Martin Mwalukanga, Diploma in Clinical Medicine; Mwangelwa Mubiana-Mbewe, BSc, MBChB, MMed, MBA; Helen B. Bwalya, BPharm.

**ZIMBABWE.** *Harare Family Care CRS*, *Harare*: Tichaona Vhembo, MBChB; Nyasha Mufukari, BPharm. *Seke North CRS*, *Chitungwiza*: Lynda Stranix-Chibanda, MBChB, MMed; Teacler Nematadzira, MBChB, MSc; Gift Chareka, MSc. *St. Mary's CRS*, *Chitungwiza*: Jean Dimairo, BPharm, Tsungai Chipato MBChB, FRCOG, MSc; Bangani Kusakara, MBChB; Mercy Mutambanengwe, BPharm; Emmie Marote, SRN MA.

## Author Contributions

**Conceptualization:** Lynda Stranix-Chibanda, Renee Browning, Mary Glenn Fowler, George K. Siberry.

**Data curation:** Lynda Stranix-Chibanda, Camlin Tierney, Jim Aizire, Godwin Chipoka, Macpherson Mallewa, Megeshinee Naidoo, Teacler Nematadzira, Bangani Kusakara, Avy Violari, Tapiwa Mbengeranwa, Boniface Njau, Lee Fairlie, Gerard Theron, Mwangelwa Mubiana-Mbewe, Sandhya Khadse, Mary Glenn Fowler.

**Formal analysis:** Lynda Stranix-Chibanda, Camlin Tierney, Mauricio Pinilla, Mary Glenn Fowler, George K. Siberry.

**Funding acquisition:** Renee Browning, George K. Siberry.

**Investigation:** Lynda Stranix-Chibanda.

**Methodology:** Lynda Stranix-Chibanda, Camlin Tierney, Renee Browning, Mary Glenn Fowler, George K. Siberry.

**Project administration:** Lynda Stranix-Chibanda, Kathleen George, Renee Browning, Mary Glenn Fowler, George K. Siberry.

**Supervision:** Lynda Stranix-Chibanda, Camlin Tierney, Mary Glenn Fowler, George K. Siberry.

**Writing – original draft:** Lynda Stranix-Chibanda, Camlin Tierney, Mary Glenn Fowler, George K. Siberry.

**Writing – review & editing:** Mauricio Pinilla, Kathleen George, Jim Aizire, Godwin Chipoka, Macpherson Mallewa, Megeshinee Naidoo, Teacler Nematadzira, Bangani Kusakara, Avy Violari, Tapiwa Mbengeranwa, Boniface Njau, Lee Fairlie, Gerard Theron, Mwangelwa Mubiana-Mbewe, Sandhya Khadse, Renee Browning, Mary Glenn Fowler, George K. Siberry.

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
