## [Decision Letter · Decision Letter 0]

9 Apr 2021

PONE-D-21-06961

Effect on growth of exposure to maternal antiretroviral therapy in breastmilk versus extended infant nevirapine prophylaxis among HIV-exposed perinatally uninfected infants in the PROMISE randomized trial

PLOS ONE

Dear Dr. Stranix-Chibanda,

Thank you for submitting your manuscript to PLOS ONE. After careful consideration, we feel that it has merit but does not fully meet PLOS ONE’s publication criteria as it currently stands. Therefore, we invite you to submit a revised version of the manuscript that addresses the points raised during the review process.

You will see that the Referees found your work of some interest. However, they also raised major criticisms. Please respond to all the comments by Reviewers 1 and 2.

We look forward to receiving your revised manuscript.

Kind regards,

Giuseppe Vittorio De Socio, MD, PhD

Academic Editor

PLOS ONE

Journal Requirements:

Please provide the names of the 14 health facility-based research sites related to the study.

We note that you have indicated that data from this study are available upon request. PLOS only allows data to be available upon request if there are legal or ethical restrictions on sharing data publicly. For information on unacceptable data access restrictions, please see http://journals.plos.org/plosone/s/data-availability#loc-unacceptable-data-access-restrictions.

3a) If there are ethical or legal restrictions on sharing a de-identified data set, please explain them in detail (e.g., data contain potentially identifying or sensitive patient information) and who has imposed them (e.g., an ethics committee). Please also provide contact information for a data access committee, ethics committee, or other institutional body to which data requests may be sent.

3b) If there are no restrictions, please upload the minimal anonymized data set necessary to replicate your study findings as either Supporting Information files or to a stable, public repository and provide us with the relevant URLs, DOIs, or accession numbers. Please see http://www.bmj.com/content/340/bmj.c181.long for guidelines on how to de-identify and prepare clinical data for publication. For a list of acceptable repositories, please see http://journals.plos.org/plosone/s/data-availability#loc-recommended-repositories.

Please include captions for your Supporting Information files at the end of your manuscript, and update any in-text citations to match accordingly. Please see our Supporting Information guidelines for more information: http://journals.plos.org/plosone/s/supporting-information.

Reviewers' comments:

Reviewer's Responses to Questions

**Comments to the Author**

1. Is the manuscript technically sound, and do the data support the conclusions?

Reviewer #1: Yes

Reviewer #2: Yes

2. Has the statistical analysis been performed appropriately and rigorously? 

Reviewer #1: No

Reviewer #2: Yes

3. Have the authors made all data underlying the findings in their manuscript fully available?

Reviewer #1: No

Reviewer #2: Yes

4. Is the manuscript presented in an intelligible fashion and written in standard English?

Reviewer #1: Yes

Reviewer #2: Yes

5. Review Comments to the Author

Reviewer #1: Abstract

- report the exact p-value instead of p-value≥0.16 in line 65 and p-value≥0.09 in line 67 and anywhere else in the manuscript, unless p-value is less than 0.001 or greater than 0.999.

Statistical methods

- t-tests to compare z-scores doesn't seem appropriate here; why not conduct an analysis of covariance (ANCOVA) by fitting a linear regression model of the post-randomisation z-scores adjusting for baseline/pre-randomisation z-scores and with treatment group-time interaction to estimate how the z-scores differ between groups at the various post-intervention times when they were measured. A likelihood ratio p-value for the treatment-time interaction would assess evidence for whether the treatments differ at the various time points - this would be a statistically more efficient way to analyse these data.

Results

- rather than presenting means and confidence intervals, I think table 2 should present means and SEs for each treatment group, followed by the difference, confidence intervals and p-values, for each outcome and time.

- actual p-values should be reported all through unless less than 0.001 or greater than 0.999; for example for p≥0.14 in line 323 and p≤0.03 in line 329

Reviewer #2: The article is generally well written with impressive design and study size. The article is generally well analyzed and presented. Except some few sentences I generally agree with the interpretations and summaries.

The conclusion in the abstract is emphasizing stunting and underweight “differences”. Based on the data you present, there seems to be no difference at week 0, week 10, week 74 and week 104 (and no clinically relevant difference at week 26). I highly doubt the 0.9% “difference” at 26 weeks which is not seen earlier or later is causally related to the intervention/drugs. Rather, I find it more probable to assume that this is due to either slight measurement inaccuracies or stochastic effects in timing of growth (as growth velocity is not constant). Thus, I suggest the conclusion in the abstract is modified, e.g. as following:

“In HEUs, growth effects from postnatal exposure to mART compared to iNVP were comparable for measures on length, weight and head-circumference with no clinically relevant differences between the groups. “

Similarly, with so comparable mean WAZ-measures between the groups, I am also skeptical to the interpretation of difference for underweight. This could be e.g. a threshold effect. If you check two additional thresholds (e.g. -1 and -3) and find similar differences, I would be a little less skeptical to the validity of these differences. As there are multiple comparisons and the study have considerable power and size, you could consider using a p<0.01 significance threshold. More importantly, I would recommend that you emphasize clinically relevant differences rather than statistically significant differences. Thus, the subsequent sentences in the conclusion of the abstract should be aligned with the manuscript, and I would slightly modified the results section accordingly.

I think the first sentence of the discussion and the conclusion in the main manuscript is more balanced.

Except these essential changes, I find the article excellent and well worth publishing.

6. PLOS authors have the option to publish the peer review history of their article (what does this mean?). If published, this will include your full peer review and any attached files.

Reviewer #1: No

Reviewer #2: **Yes: **Lars T. Fadnes

---

## [Author Response · Author response to Decision Letter 0]

11 May 2021

REVIEWER #1: 

R1.1 - Abstract - report the exact p-value instead of p-value≥0.16 in line 65 and p-value≥0.09 in line 67 and anywhere else in the manuscript, unless p-value is less than 0.001 or greater than 0.999.

Thank you for the comment. We have revised the Abstract [see lines 66 and 68] to show that the several p-values presented there were all >=0.16 and >=0.09, respectively, and trust that this adequately addresses the concern raised. 

R1.2 - Statistical methods - t-tests to compare z-scores doesn't seem appropriate here; why not conduct an analysis of covariance (ANCOVA) by fitting a linear regression model of the post-randomisation z-scores adjusting for baseline/pre-randomisation z-scores and with treatment group-time interaction to estimate how the z-scores differ between groups at the various post-intervention times when they were measured. A likelihood ratio p-value for the treatment-time interaction would assess evidence for whether the treatments differ at the various time points - this would be a statistically more efficient way to analyse these data.

We understand the approach suggested by Reviewer 1, however, the protocol-specified primary objective was to conduct a cross sectional comparison at postpartum Week 26, with the primary outcome measure of LAZ. Other timepoints and measures were secondary. After discussion by the writing team, we decided to clarify the statistical analysis approach in the revised text [see Statistical Methods lines 221-225 and Abstract line 57] and believe that the explanation is now clear. 

R1.3 - Results - rather than presenting means and confidence intervals, I think table 2 should present means and SEs for each treatment group, followed by the difference, confidence intervals and p-values, for each outcome and time.

Table 2 has been adjusted in line with Reviewer 1’s comment and week 0 infant anthropometric measures were moved to Table 1 [see lines 300-309 in the revised manuscript]. Please note that we took the opportunity to correct an erroneous percentage for stunting in Table 2. The text preceding the table, table title, and footnote were updated accordingly. Table 1 was updated thus: addition of week 0 LAZ, WAZ, and HCAZ under infant characteristics (including % z score < -2); removal of rows for “N” for maternal characteristics (since they are redundant given the column headers and # missing); addition of 10%-90% for infant characteristics (to be consistent with the descriptive statistics presented for mothers); footnote updated accordingly [see lines 271-279].

R1.4 - Results - actual p-values should be reported all through unless less than 0.001 or greater than 0.999; for example for p≥0.14 in line 323 and p≤0.03 in line 329

We have attempted to address this concern by clarifying that we present several p-values in the examples mentioned where no actual p-value is shown. Please see the inserted text in the revised manuscript [for example in line 333]. 

REVIEWER #2: 

R2.0 - The article is generally well written with impressive design and study size. The article is generally well analyzed and presented. Except some few sentences I generally agree with the interpretations and summaries.

Thank you for the comment.

R2.1 - The conclusion in the abstract is emphasizing stunting and underweight “differences”. Based on the data you present, there seems to be no difference at week 0, week 10, week 74 and week 104 (and no clinically relevant difference at week 26). I highly doubt the 0.9% “difference” at 26 weeks which is not seen earlier or later is causally related to the intervention/drugs. Rather, I find it more probable to assume that this is due to either slight measurement inaccuracies or stochastic effects in timing of growth (as growth velocity is not constant). Thus, I suggest the conclusion in the abstract is modified, e.g. as following:

“In HEUs, growth effects from postnatal exposure to mART compared to iNVP were comparable for measures on length, weight and head-circumference with no clinically relevant differences between the groups. “

Thank you for the insight and the suggested revision which was considered by the writing team. We opted to use the sentence you provided for the Conclusion in the Abstract [see lines 71-82 in the revised manuscript].

R2.2A - Similarly, with so comparable mean WAZ-measures between the groups, I am also skeptical to the interpretation of difference for underweight. This could be e.g. a threshold effect. If you check two additional thresholds (e.g. -1 and -3) and find similar differences, I would be a little less skeptical to the validity of these differences.

We hope the adjusted interpretation presented in the revised manuscript likely addresses this concern. To explain the -2 SD threshold selection, that is the standard clinical definition for stunting and underweight based on World Health Organisation definitions and is not driven by study data. Thus, we made minor clarifications to the Methods and inserted new reference #21 to illustrate the basis for that threshold in the revised manuscript [see lines 237—239]. However, we did carry out a sensitivity analysis around -2, with thresholds of -1.9 and -2.1 to address potential random noise right around this measurement that could result in disproportionate assignments to one category. Results were very similar, and we determined there was no need to include them in the manuscript.

R2.2B - As there are multiple comparisons and the study have considerable power and size, you could consider using a p<0.01 significance threshold. 

Indeed, this study does have considerable size. The 0.05 significance level was pre-specified in our analysis plan.

R2.2C - More importantly, I would recommend that you emphasize clinically relevant differences rather than statistically significant differences. Thus, the subsequent sentences in the conclusion of the abstract should be aligned with the manuscript, and I would slightly modified the results section accordingly.

We agree with Reviewer 2’s helpful guidance. Accordingly, we revised the Conclusions in the Abstract [see lines 71-82] and removed the term ‘significant’ from several sentences in the Results section [see lines 314-346] to remove any misrepresentation of clinical relevance. We do include p-values in Table 2, as suggested by Reviewer 1 (comment R1.3), but they are not emphasized heavily in the text.

R2.2D - I think the first sentence of the discussion and the conclusion in the main manuscript is more balanced.

Except these essential changes, I find the article excellent and well worth publishing.

Thank you for the insightful comments and suggestions which we have taken on board and used to strengthen the manuscript we now resubmit.

EDITOR'S COMMENTS

These have been attended to within the cover letter, as instructed.

There are no supplemental files for this submission -- the protocol and COSORT checklist were submitted as supplemental files after the initial package was uploaded following guidance.

---

## [Decision Letter · Decision Letter 1]

8 Jun 2021

PONE-D-21-06961R1

Effect on growth of exposure to maternal antiretroviral therapy in breastmilk versus extended infant nevirapine prophylaxis among HIV-exposed perinatally uninfected infants in the PROMISE randomized trial

PLOS ONE

Dear Dr. Stranix-Chibanda,

Thank you for submitting your manuscript to PLOS ONE. After careful consideration, we feel that it has merit but does not fully meet PLOS ONE’s publication criteria as it currently stands. Therefore, we invite you to submit a revised version of the manuscript that addresses the points raised during the review process.

We invite you to submit a revised version of the manuscript that addresses  minor suggestions by Reviewer #1 and #2.

We look forward to receiving your revised manuscript.

Kind regards,

Giuseppe Vittorio De Socio, MD, PhD

Academic Editor

PLOS ONE

Journal Requirements:

Reviewers' comments:

Reviewer's Responses to Questions

**Comments to the Author**

1. If the authors have adequately addressed your comments raised in a previous round of review and you feel that this manuscript is now acceptable for publication, you may indicate that here to bypass the “Comments to the Author” section, enter your conflict of interest statement in the “Confidential to Editor” section, and submit your "Accept" recommendation.

Reviewer #1: All comments have been addressed

Reviewer #2: (No Response)

2. Is the manuscript technically sound, and do the data support the conclusions?

Reviewer #1: Yes

Reviewer #2: Yes

3. Has the statistical analysis been performed appropriately and rigorously? 

Reviewer #1: Yes

Reviewer #2: Yes

4. Have the authors made all data underlying the findings in their manuscript fully available?

Reviewer #1: No

Reviewer #2: (No Response)

5. Is the manuscript presented in an intelligible fashion and written in standard English?

Reviewer #1: Yes

Reviewer #2: (No Response)

6. Review Comments to the Author

Reviewer #1: Please correct the typographical error at the top of page 11 of 24 - you mean Kruskal-Wallis, not Kruskal-Wallace.

Reviewer #2: I think the article has improved and that the first response is fine. I do not fully agree with the approaches on question 2.2, but that being said the discussion of that finding and conclusion still look generally fine. I am aware that z-score -2 is a commonly used threshold for stunting and underweight, but e.g. -3 is also a commonly used threshold for severe underweight/stunting. See also the article by Briend et al. on arguments for not only focusing on a single threshold (Briend A, Van den Broeck J, Fadnes LT: Target weight gain for moderately wasted children during supplementation interventions - a population-based approach. Public health nutrition 2011:1-7.) I do not see the option of adding sensitivity analysis for z-score thresholds of -1.9 and -2.1 (which is very close to the original analysis) as good alternatives to what I suggested of a sensitivity analyses with z-score thresholds of -3 and -1. Unless you see similar results with thresholds of -3 and -1, I think the presentation of differences in the results section is not optimal. However, if you show similar trends for sensitivity analyses, you will probably convince me of your presentation/interpretation. For the results section as I consider the confidence intervals for the dichotomous outcomes are relatively wide (stunting/overweight), I suggest sensitivity analysis with thresholds further away from the original analysis (and that these preferable are added as a supplementary table) to guide whether these are threshold effects. If so, that could be mentioned shortly in the results/discussion section. Also a note relating to the p-value threshold, even though p<0.05 was predefined, it is still possible to guide interpretation not only dichotomously but to discuss degree of association (and chance of random associations versus causal inferences).

That being said, the study and article is generally very good and well worth publishing. I just think adding a sensitivity analysis which is more different from the original analysis would lift the paper even more.

7. PLOS authors have the option to publish the peer review history of their article (what does this mean?). If published, this will include your full peer review and any attached files.

Reviewer #1: No

Reviewer #2: **Yes: **Lars Fadnes

---

## [Author Response · Author response to Decision Letter 1]

30 Jun 2021

REVIEWER #1: 

[Revision 1] R1.1 - Please correct the typographical error at the top of page 11 of 24 - you mean Kruskal-Wallis, not Kruskal-Wallace.

Thank you for identifying this error which is corrected in line 228 of the marked-up revised manuscript [Revision 2]. 

REVIEWER #2: 

[Revision 1] R2.0 - I think the article has improved and that the first response is fine. 

Noted.

[Revision 1] R2.1 - I do not fully agree with the approaches on question 2.2, but that being said the discussion of that finding and conclusion still look generally fine. I am aware that z-score -2 is a commonly used threshold for stunting and underweight, but e.g. -3 is also a commonly used threshold for severe underweight/stunting. See also the article by Briend et al. on arguments for not only focusing on a single threshold (Briend A, Van den Broeck J, Fadnes LT: Target weight gain for moderately wasted children during supplementation interventions - a population-based approach. Public health nutrition 2011:1-7.) I do not see the option of adding sensitivity analysis for z-score thresholds of -1.9 and -2.1 (which is very close to the original analysis) as good alternatives to what I suggested of a sensitivity analyses with z-score thresholds of -3 and -1. Unless you see similar results with thresholds of -3 and -1, I think the presentation of differences in the results section is not optimal. However, if you show similar trends for sensitivity analyses, you will probably convince me of your presentation/interpretation. For the results section as I consider the confidence intervals for the dichotomous outcomes are relatively wide (stunting/overweight), I suggest sensitivity analysis with thresholds further away from the original analysis (and that these preferable are added as a supplementary table) to guide whether these are threshold effects. If so, that could be mentioned shortly in the results/discussion section. 

The writing team accepted the reviewer’s comment and are grateful for the reference provided. We agree with not focusing on a single threshold for this secondary outcome measure. We conducted additional sensitivity analyses using thresholds further away from the original z-score threshold (-2.0). For length-for-age (LAZ), the thresholds used were -1.0 and -3.0, as suggested. However, the number of infants with weight-for-age (WAZ) below -3.0 was very low (<2%) at all time points, therefore the thresholds used for WAZ were -1.0 and -2.5. The direction of the odds ratio estimate remains consistent across the different thresholds for both LAZ and WAZ. The results of this sensitivity analysis are now mentioned in line 333 of the marked-up revised manuscript [Revision 2]. The sensitivity analysis results are uploaded with this rebuttal letter for review. We have included a table as new Supporting Information for the revised manuscript (caption inserted at the end of the manuscript document), and include figures below illustrating additional thresholds for the LAZ and WAZ comparisons (Figure A and Figure B - uploaded in the separate document Response to Reviewers). 

[Revision 1] R2.2 - Also a note relating to the p-value threshold, even though p<0.05 was predefined, it is still possible to guide interpretation not only dichotomously but to discuss degree of association (and chance of random associations versus causal inferences).

That being said, the study and article is generally very good and well worth publishing. I just think adding a sensitivity analysis which is more different from the original analysis would lift the paper even more.

We reviewed the manuscript with this note in mind. In Revision 2, we removed “significant” from the Abstract and Results in lines 64 and 304, and p-values in lines 70 and 325. Upon review, we revised lines 359-360 in the Discussion that appeared to rely too heavily on the p-values. 

Figure A. Sensitivity of odds ratio estimate to varying length-for-age Z-score (LAZ) binary categorization

Figure B. Sensitivity of odds ratio estimate to varying weight-for-age Z-score (WAZ) binary categorization

---

## [Editor Report · Decision Letter 2]

13 Jul 2021

Effect on growth of exposure to maternal antiretroviral therapy in breastmilk versus extended infant nevirapine prophylaxis among HIV-exposed perinatally uninfected infants in the PROMISE randomized trial

PONE-D-21-06961R2

Dear Dr. Stranix-Chibanda,

We’re pleased to inform you that your manuscript has been judged scientifically suitable for publication and will be formally accepted for publication once it meets all outstanding technical requirements.

Kind regards,

Giuseppe Vittorio De Socio, MD, PhD

Academic Editor

PLOS ONE
---

## [Editor Report · Acceptance letter]

4 Aug 2021

PONE-D-21-06961R2 

Effect on growth of exposure to maternal antiretroviral therapy in breastmilk versus extended infant nevirapine prophylaxis among HIV-exposed perinatally uninfected infants in the PROMISE randomized trial 

Dear Dr. Stranix-Chibanda:

I'm pleased to inform you that your manuscript has been deemed suitable for publication in PLOS ONE. Congratulations! Your manuscript is now with our production department. 

Kind regards, 

on behalf of

Dr. Giuseppe Vittorio De Socio 

Academic Editor

PLOS ONE